# A programmable hybrid digital chemical information processor based on the Belousov-Zhabotinsky reaction

Abhishek Sharma[1], Marcus Tze-Kiat Ng [1], Juan Manuel Parrilla Gutierrez[1], Yibin Jiang [1] & Leroy Cronin [1] ✉

The exponential growth of the power of modern digital computers is based upon the miniaturization of vast nanoscale arrays of electronic switches, but this will be eventually constrained by fabrication limits and power dissipation. Chemical processes have the potential to scale beyond these limits by performing computations through chemical reactions, yet the lack of well-defined programmability limits their scalability and performance. Here, we present a hybrid digitally programmable chemical array as a probabilistic computational machine that uses chemical oscillators using Belousov-Zhabotinsky reaction partitioned in interconnected cells as a computational substrate. This hybrid architecture performs efficient computation by distributing information between chemical and digital domains together with inbuilt error correction logic. The efficiency is gained by combining digital logic with probabilistic chemical logic based on nearest neighbour interactions and hysteresis effects. We demonstrated the computational capabilities of our hybrid processor by implementing one- and two-dimensional Chemical Cellular Automata demonstrating emergent dynamics of life-like entities called Chemits. Additionally, we demonstrate hybrid probabilistic logic as a viable logic for solving combinatorial optimization problems.

The exponential increase in computing power has been driven by the vast growth of transistors on silicon chips[1]. This growth was made possible by developments in fabrication technology reducing the feature sizes of the transistors[2], but this paradigm is currently approaching the limits imposed by physics as quantum effects become more pronounced[3]. To overcome these limitations, various novel computational architectures based on physical and chemical processes have been proposed[4–8]. Quantum computers show the potential to solve problems that are intractable on classic computing machines but currently suffer from scalability issues due to error correction[9]. Alternative computing substrates and unconventional computation paradigms are being developed based on mapping computational logic to various physical phenomena[10]. These tend to emulate transistor-based logic gates and other circuit components into the physical domain using architectures based on Boolean circuits[11] or

discovered using artificial intelligence[12]. Other classic computational architectures which utilize the true nature of physical phenomena include reaction-diffusion[13] and neuromorphic computers[14]. These architectures and their algorithms have been designed to solve a specific set of abstract mathematical problems[15–18]. The challenge is to develop new platforms that can take advantage of chemical substrates but can be easily programmable.

The key concept of hybrid electronic–chemical computation is inspired by the previously developed concepts of heterotic[19] as well as physical computation[20] where an amalgamation of different computational systems such as digital, chemical, optical, etc., are used together for efficient computation. Herein, we present a probabilistic information processor based on a hybrid electronic–chemical computational architecture that is digitally addressable which utilizes both analogue and their digital-equivalent states for information

[1]School of Chemistry, The University of Glasgow, University Avenue, Glasgow G12 8QQ, UK. ✉e-mail: Lee.Cronin@glasgow.ac.uk

processing. The efficiency is gained by distributing the computational task into different computational substrates which are efficient in performing specific operations using their natural propensity. This device is built from a programmable chemical array-based[21] around a new type of hybrid chemical state machine for information processing. The hybrid computational array operates using the Belousov−Zhabotinsky (BZ) reaction, which is an oscillating chemical reaction[22,23]. The cells are arranged in a rectangular array and are individually programmable with a central oscillator driver and have four tuneable gates that allow the oscillations to be coupled between adjacent cells in the grid, which defines the basic computational element. By programming cell and interfacial controls, the emerging chemical oscillations of the $i$th cell (together with neighbouring cells defined with index $j$) are defined by the analogue chemical state ($\underline{CS}_i^t$) that can be monitored and mapped to a digitally representable chemical state ($CS_i^t$) for each cell at time step $t$. The chemical state evolution in the analogue domain occurs via physical processes such as chemical oscillations, while the digital state is based on a finite state logic. Both the analogue and digital representations of chemical states are synchronized using a clock signal created by the chemical oscillations. Thus, this approach uses a hybrid electronic−chemical logic where the digital domain applies deterministic logic using finite state machine implemented on digital states ($CS_i^t$) and the chemical domain performs analogue computation using interacting chemical oscillations. A simplified representation of the time evolution of the hybrid electronic−chemical logic in the analogue domain using chemical states ($\underline{CS}_i^t$) can be defined by

$$\underline{CS_i^{t+1}} = G\left(F\left(\underline{CS_i^t}, \underline{CS_j^t}\right), \underline{CS_i^t}\right) \tag{1}$$

where $F$ represents the digital operation based on the readout of the chemical states and $G$ represents interactions in the physical system. $F$ can be defined as a digital finite state machine (FSM) that reads the current analogue signal from the interacting physical system between nearest neighbours and performs operations back into the analogue domain and $\underline{CS_j^t}$ represents chemical states of neighbouring cells of the primary cell with chemical state $\underline{CS_i^t}$. The physical system evolves in a high dimensional space controlled by the complex myriad interactions between the chemical oscillations and their hysteresis effects together with the effect of localized digital operations and leads to the probabilistic outcomes of the emerging chemical states. Hence, the computation is performed by iterating single-step operations involving analogue and digital states with finite state logic defined for a specific problem.

In our experiments, using a digitally programmable input−output (I/O) system, we showed that it is possible to have an error correction system whereby the oscillatory chemical state can be reinforced, and the effects of phase shift can be eliminated so that the hybrid electronic control system amplifies and stabilizes the chemical states (see Fig. 1). As a proof of programmability and computation, we showed that the system could embody a hybrid electronic−chemical form of cellular automaton (CA) and increase the connectivity in the configuration space way beyond the digital feature space even for simple rules of the elementary CA[24]. The probabilistic nature of the hybrid computational architecture was demonstrated by implementing a two-dimensional probabilistic chemical cellular automata (CCA) which shows emergent dynamics similar to that seen in Conway's Game of Life[25]. Additionally, we also implemented solutions to combinatorial optimization problems such as number partitioning[26], Boolean satisfiability[27], and the travelling salesman problem[28] using a hybrid computational approach.

## Results
### The chemical computing platform
The chemical array exploits the excitability of the non-linear chemical oscillations of the Belousov −Zhabotinsky (BZ) reaction via localized

spatial control[22,23]. The BZ reaction is highly excitable, can be maintained far from equilibrium, and it can be both spatially and temporally addressable. To exploit these features, we designed an experimental setup where we can programme the addressable medium using an electronically controlled input system (see Fig. 1a, b). The experimental architecture consists of a 3D-printed 1D and 2D grid of interconnected reactors supported on an array of motors equipped with magnetic heads. At the centre of each reactor and the interface of neighbouring cells, magnetic stirrers are placed to match the position of the motor shaft. Each motor is individually addressable, and its speed is controlled using a pulse width modulation (PWM) signal generated using a microcontroller. The schematic diagram and the physical implementation of the two-dimensional experimental setup are shown in Fig. 2, Supplementary Information Section 1 and Supplementary Video 1. At the start of each experiment, the reagents required to initiate the BZ reaction (solutions of malonic acid, potassium bromate, an iron-based redox catalyst and sulfuric acid) are added to the reactors using an automated liquid handling system interface (see the "Methods" section and Supplementary Information Section 1). The role of the central cell stirrer is to initiate and then maintain the chemical oscillations and their amplitude in a programmable way by varying the stirring speed. The mass transfer due to the hydrodynamic coupling between the neighbouring cells leads to interactions between the chemical oscillations of the BZ reactions (see Fig. 1c, d) and Supplementary Video 2. By tuning the speed of the interfacial stirrers (PWM levels), we can control and programme the strength of intercellular couplings between two interacting cells and limit them to their nearest neighbours.

The BZ oscillations induced in a single cell are extremely sensitive to the composition of local redox species and the time of the actuation of the stirrer. This extreme sensitivity to the initial state, localized fluctuations, and time of actuation causes the phase of the chemical oscillations in individual cells to show significant drift with time. We observed that these unfavourable phase shifts between individual cells potentially limited the programmability of the system, and an error correction process to prevent decoherence was needed. To develop a hybrid computational logic, we investigated the phenomenological behaviour of oscillations in single and coupled neighbouring cells. The BZ system combined with the stirrer acts like a forced-damped oscillator. Chemical oscillations appear when the cell stirrer is active (forced oscillator) and once the stirrer is deactivated, the amplitude of oscillations starts falling (damped oscillator) and disappears completely after a few cycles. In the case of coupled neighbouring cells, no interaction occurs between them when the interfacial stirrer is off. On activating the interfacial stirrer, bidirectional interaction occurs and weakly coupled neighbouring cell oscillations come in phase independent of their initial phase differences. Additionally, the interactions in both one- and two-dimensional geometry are confined to nearest neighbours only (see Supplementary Information Section 2). In the digital domain, for the finite state logic, two chemical states were defined as $CS_i^t = 0$ or $CS_i^t = 1$. To eradicate the errors in defining digital states due to the strong hysteresis effects in the chemical domain, the chemical state $CS_i^t = 0$ was defined by low-amplitude chemical oscillations instead of no oscillations. The low oscillation modes were generated by stirrers in pulsing mode. The high amplitude chemical oscillation was represented in the digital domain as $CS_i^t = 1$ and was created in the chemical domain by continuous mode at a high stirring rate. Additionally, to address the potential for errors resulting from state-decoherence (phase shift) in the oscillations among the cells, we found that the introduction of a global 'clock' signal (SYNC) was possible by creating weak coupling between neighbouring cells via interfacial stirrers. The weak coupling keeps chemical oscillations over all the cells synchronized (see Supplementary Information Section 2). These global weak oscillations are used for defining clocking signals which allows us to prevent unwanted dephasing (see Fig. 3).

As the BZ reaction proceeds and the malonic acid 'fuel' is consumed, the amplitudes of the oscillations decrease with time (see Fig. 3a, b). The deviation between the oscillations in the interconnected network of cells is shown in Fig. 3c. To consistently define discrete chemical states based on the observed colour amplitudes, we trained a convolutional neural network (CNN) on a dataset of time-dependent images labelled with discrete states. The three discrete states based on time-dependent colour classification are red (R), light blue (LB) and blue (B) which are also referred to as CNN states. In the presence of only weak oscillations, two distinct CNN states (Red and Light Blue) are recognized, while in the presence of strong oscillations, three distinct states (Red, Light Blue and Blue) emerge (see Fig. 3d), and we use them to describe the pattern of oscillations in a finite state machine called as rFSM (recognition finite state machine) to determine the chemical states. An interpreted simplified example of clock states corresponding to Fig. 3d is shown in Fig. 3e (see Supplementary Information Section 4 for actual implementation). In our current design, the rFSM logic consists of two different digital states: $CS_i^t = 0$

and $CS_i^t = 1$. The emergence of state $CS_i^t = 0$ occurs when a weak wave pattern is observed ($R \rightarrow LB \rightarrow R$) and the emergence of state $CS_i^t = 1$ occurs when a stronger oscillation is observed ($R \rightarrow LB \rightarrow B \rightarrow LB \rightarrow R$). From the chemical state readout, a feedback loop can be completed by implementing a deterministic digital state machine ($D$) that reads out the chemical states of all or a subset of cells as inputs and returns the PWM levels of the stirrers. These new PWM levels were then applied to the cell and interfacial stirrers (see Supplementary Information Sections 3 and 4 for the implementation of rFSM).

### Probabilistic logic and one-dimensional chemical cellular automata (1D-CCA)

A single information processing loop can be represented by a combination of four state machines ($C,T,D,P$) acting on digital ($CS_i^t$) and an analogue representation of chemical states ($\underline{CS_i^t}$) (see Fig. 4a and b). The state machine $C$ is probabilistic and represents the evolution of $\underline{CS_i^t}$ while $D$ represents a deterministic digital state machine that reads

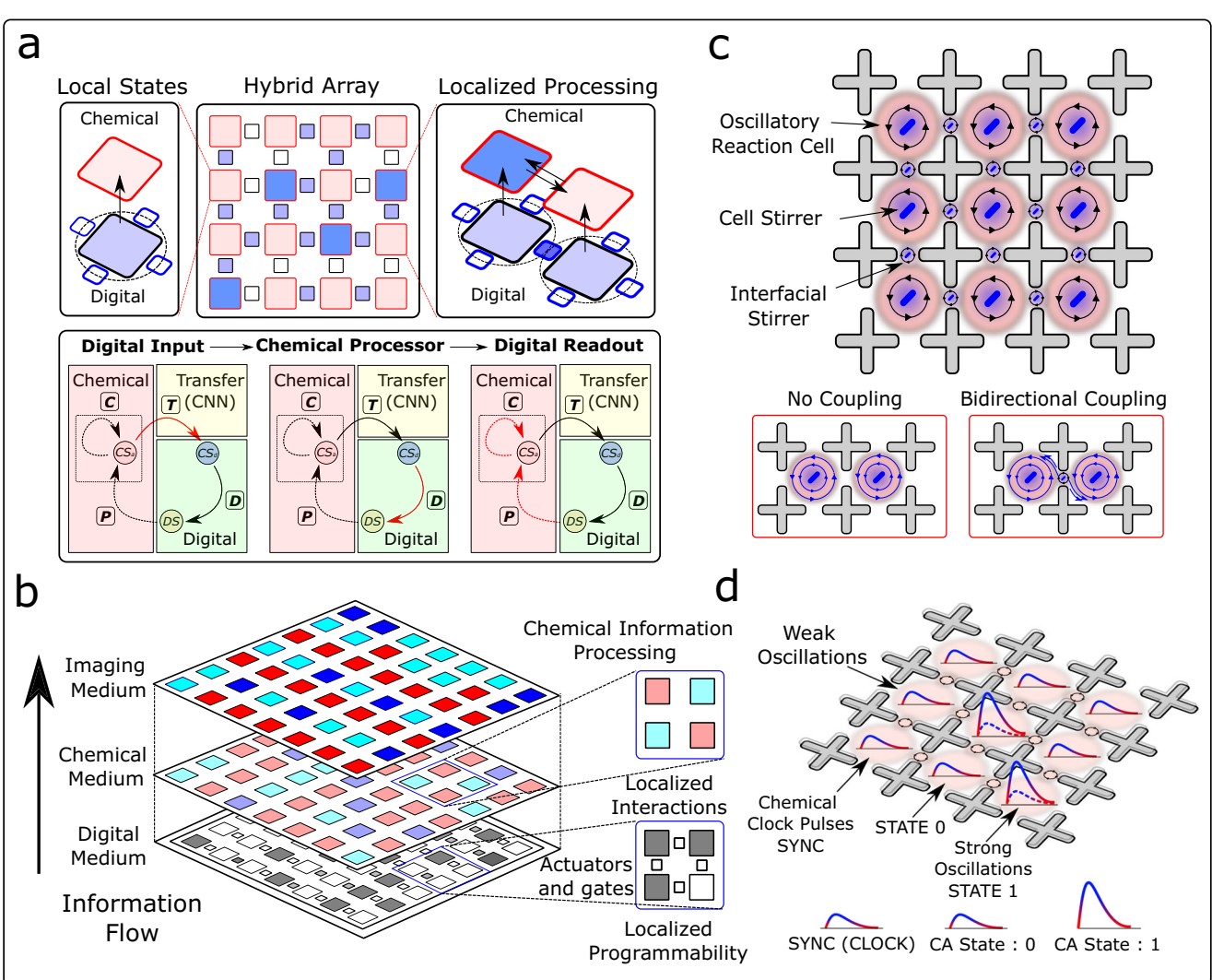

**Fig. 1 | Conceptual and schematic design. a** Conceptual diagram of the proposed hybrid computational architecture comprising an array of chemical reactors. The top figure shows hybrid digital–chemical information processing within single and coupled hybrid logical units. The bottom figure demonstrates a single processing step of the hybrid state machine with information looping between digital and chemical domains. Here, *D, P, C & T* represent digital, physical, chemical and transfer state machines (with $CS_a \equiv \underline{CS_i^t}$ and $CS_d \equiv CS_i^t$). **b** Shows a pictorial representation of how the information propagates together with local information processing in the chemical array. The weakly connected network in the chemical medium provides the global clock (SYNC signal) on which the local interactions process information and perform computational operations. **c** and **d** Schematic diagram of the two-dimensional Belousov–Zhabotinsky (BZ)-based hybrid computation architecture showing how local cellular vortices interact by tuning the speed of the interfacial stirrers. The amplitude of the oscillations is controlled by the speed of the cell stirrers and can be used to define discrete states for information processing. Due to the well-defined periodic behaviour in the weak coupling limit, these oscillations can also be used to create a global clocking SYNC signal for decision-making.

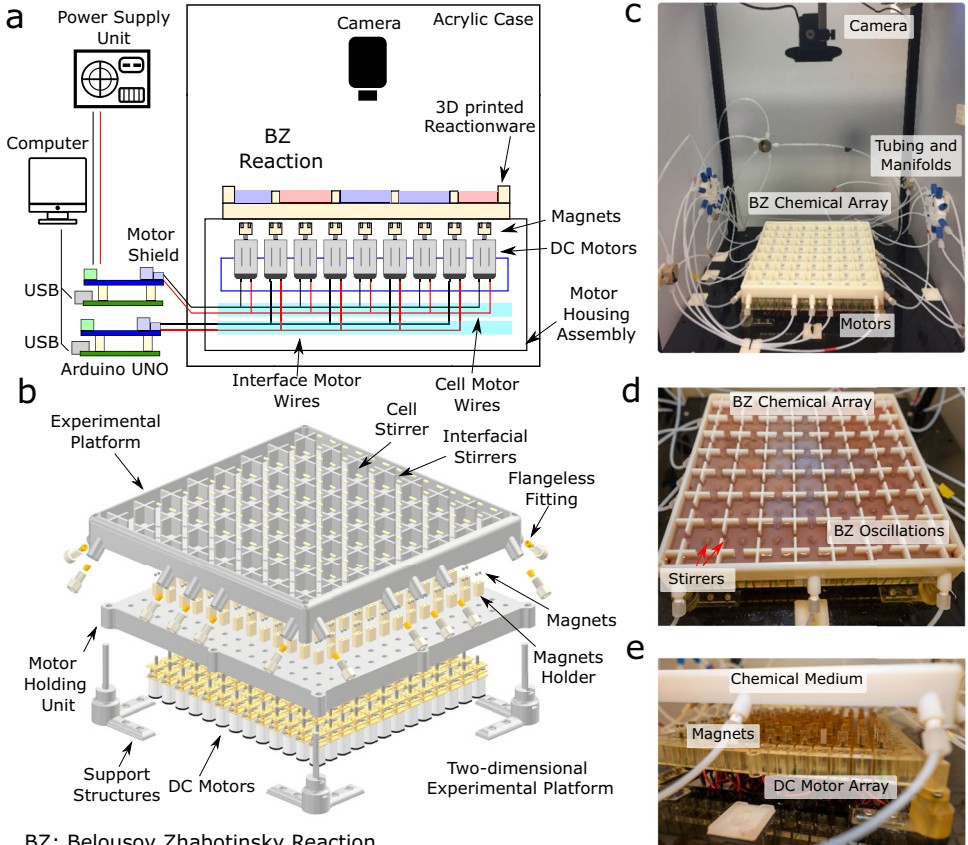

**Fig. 2 | Schematic design and physical implementation of the experimental platform. a** Schematic diagram of the automated closed-loop experimental setup showing a 3D printed reactor where the Belousov–Zhabotinsky reaction (BZ) occurs, motor control, imaging unit and supporting electronics. **b** Exploded view of the 3d printed reactor array with stirrers mapped with motor array and fluidic connections. **c** Complete experimental setup inside a light-controlled acrylic housing with fluidic connections from the pump control unit. **d** Closeup view of the 3D printed reactor array with emerging chemical oscillation patterns. **e** Closeup view of the motor array with magnets connected to motor shafts controlling stirrer actuation. See Supplementary Information Section 1 for complete details.

$CS_i^t$ and updates stirrer states (DS). The state machines $T$ and $P$ represent analogue to digital chemical state conversion ($CS_i^t = T(\underline{CS_i^t})$) and the physical effects of stirrers represent digital to analogue chemical states, respectively. The time evolution of the digital and analogue representation of chemical states in hybrid probabilistic computation can be represented by

$$CS_i^t = K^{t-1} K^{t-2} \dots K^1 K^0(C(\text{IC}))$$
$$K^t(CS_i^t) \equiv C\left(P\left(D\left(CS_i^t, CS_j^t\right)\right), \underline{CS_i^t}\right) \qquad (2)$$

where $K$ represents a hybrid state machine comprising of four state machines $C$, $T$, $D$, $P$ and $i, j$ represents the central and neighbouring cells and the state machine $T$ is included in $C$ for simplicity. IC defines the initial conditions. The emergence of the new chemical state has both implicit and explicit dependence on the previous chemical states. The implicit dependence comes from the hysteresis effect of the oscillations and the explicit dependence comes from the physical interaction of the stirrer state ($P$) on the oscillations which depends on the previous chemical states via finite state logic $D(CS_i^t, CS_j^t)$.

To demonstrate the working principle of the closed-loop hybrid electronic–chemical logic and the programmability with clocking logic, we implemented the elementary cellular automata (CA) rules (Rule 30, 110 and 250, see Supplementary Information Section 4) in a fully deterministic way, see Rule 30 as an example Fig. 4c and Supplementary Video 3. In the deterministic mode as there is a one-to-one mapping between stirrer (DS) and chemical states ($CS_i^t$), information loops through digital and chemical domains where the chemical state

machine mirrors the digital state machine precisely (see Fig. 4a). Next, we introduced the probabilistic computational logic by introducing hybrid automaton rules (see Fig. 4b). Consequently, the new chemical states in the analogue domain emerge probabilistically where CA rule 30 is modified by creating asymmetric actuation on interfacial stirrers as an example (see Fig. 4c) (and Supplementary Information Section 4). The probabilistic outcomes of the chemical states can be seen in a single cell with the effect of stirrer speed on the emerging peak amplitudes (see Fig. 4d). By introducing new automaton rules that exploit the probabilistic computation mode enabled by the high dimensional space associated with the chemical state machine ($C$), larger configuration spaces can be explored.

Inspired by the elementary CA rules, we also developed a family of one-dimensional chemical cellular automata (1D-CCA) rules based on a closed-loop state machine utilizing PWM and chemical states. A CCA rule can be defined on an experimental platform based on the observed chemical states and selected PWM levels of interfacial and cell stirrers. In the absence of interfacial stirrers, we can recreate elementary CA rules within a closed feedback loop due to one-to-one mapping between the PWM states of cell stirrers and observed chemical states. However, when the interfacial stirrers are active, due to hydrodynamic coupling between the neighbouring cells and coupled hysteresis effects, one-to-one mapping between PWM and chemical states does not exist and novel patterns can emerge out. As an example, we describe the 1D-CCA state machines combined with a simple phenomenological model which was then used to simulate the emergence of patterns in a one-dimensional geometry (see Supplementary Information Section 5). 1D-CCA rules which define the state

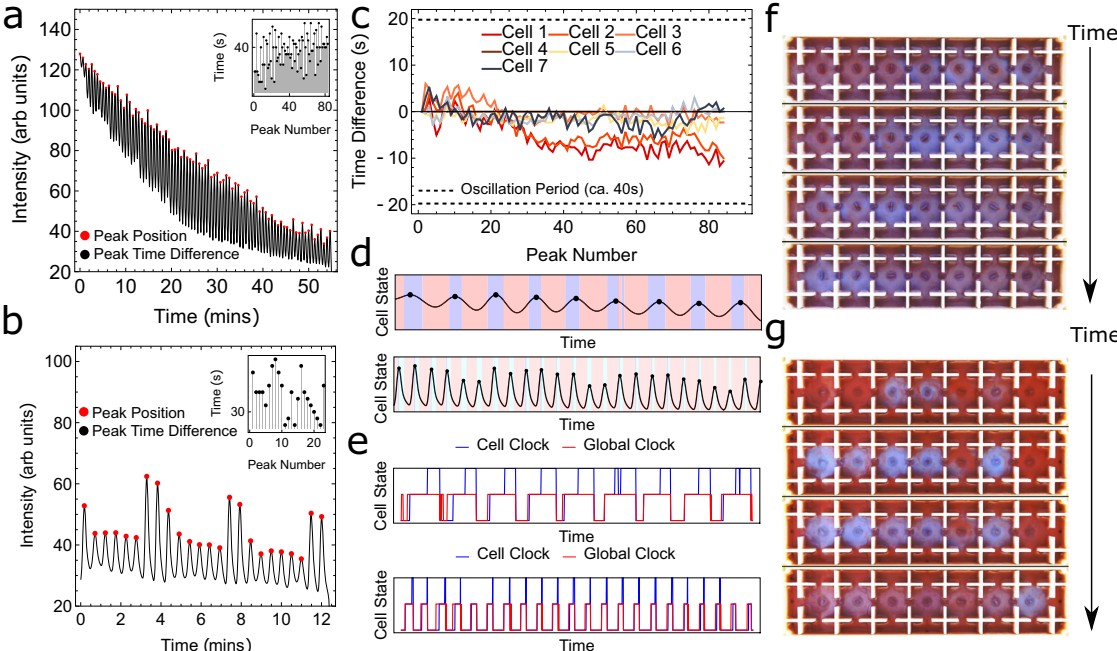

**Fig. 3 | Chemical clocking in one-dimensional hybrid computation experiments. a** Chemical oscillations were recorded for 60 min in a continuously stirred single reactor. **b** Chemical oscillations were recorded in a single cell with LOW/HIGH states with programmable switching based on a global clocking signal. The red dots in **a** and **b** are the detected peak oscillations. The black dots in the inset plot represent the time differences between the consecutive peaks at different peak numbers. **c** Deviations between the peak oscillations with time (peak number) between different interconnected cells with global clocking, **d** shows chemical oscillations and discrete chemical states by applying convolutional neural network (CNN) for a single oscillatory cell vs. time for pure clocking tests (top

figure, with red and light blue as two states) and cellular automata tests (down figure, with red, cyan, and blue shades represents CNN states "red", "light blue", and "blue"). In both cases, the black dots represent the peak position of oscillations, and **e** shows two different examples of simplified representations of cell and global clocks interpreted from CNN output (for cell clock: low, middle, and high states represent "none", "tick", and "tock" states, and for global clock: middle and low represents "tick" and "tock" states). See Supplementary Information Section 4 for complete details. **f** and **g** are the temporal snapshots of the actual experiments in a one-dimensional cell array demonstrating clocking waves without and with chemical mapped chemical states.

machine to act on stirrer based on chemical states comprised of two different values, $\{C_{rule}\}$–$\{I_{rule}\}$. $C_{rule}$ updates the central cell stirrer PWM state based on the chemical states of nearest-neighbouring cells $(C_i^t, C_{i-1}^t, C_{i+1}^t)$ similar to the elementary CA rule table. $I_{rule}$ updates the PWM states of the two interfacial stirrers based on the chemical states of the two connecting cells $(C_{i-1}^t, C_i^t)$ and $(C_i^t, C_{i+1}^t)$. See Supplementary Information Section 5 for quantification of input and chemical states, probabilistic model, and examples of 1D-CCA using a phenomenological model.

## Two-dimensional chemical cellular automata (2D-CCA)

By extending the CCA into two dimensions it is possible to explore the dynamics and emergence of complex patterns based on local rules defined by the hybrid electronic–chemical logic. Inspired by Conway's Game of Life and using a similar concept described in 1D-CCA (based on chemical states and PWM levels of interfacial and cell stirrers), as a discrete model of computation, we developed a two-dimensional probabilistic cellular automaton with extended multi-cellular life-like entities. In the 2D-CCA, we defined the basic emergent life-like units as chemical entities (Chemits), which are comprised of a combination of five nearest-neighbouring cells in the von Neumann neighbourhood (see Fig. 5a, b) for experimental and pictorial representation of the Chemit. The objective is to observe emergent dynamics of Chemits such as replication-propagation in the presence of weak fluctuating oscillations over the whole cell array. The positions and dynamics of Chemits are defined by the combination of digital and analogue representation of chemical states as described previously. We implemented a 2D-CCA digital state machine that takes the chemical states $(CS_i^t = \{0,1\})$ emerging from the probabilistic outcomes of the chemical information processing and outputs 4-state PWM logic with states

$\{S_0, S_1, S_2, S_3\}$ which defines the stirrer operations. $S_0$ corresponds to the inactive stirrer state, $S_1$ introduces the random fluctuations (weak chemical oscillations on randomly selected cells), $S_2$ creates the Chemit core and $S_3$ introduces the extended interacting body of the Chemit which leads to the interactions with the surroundings. The chemical oscillations created by PWM state $S_1$ in the absence of Chemits only leads to the $CS_i^t = 0$ chemical state and hence cannot create a new Chemit. The Chemit core defined by a high PWM value ($S_2$) creates high $CS_i^t = 1$ states. The pictorial representation of the complete closed-loop probabilistic logic of two-dimensional CCA representing the emergence of Chemits is shown in Fig. 5c.

In the experiments, based on the local rules the Chemits show emergent dynamics such as propagation, replication, and competition analogous to life-like species (see Supplementary Video 4). The emergence of a high chemical state at a specific cell occurs probabilistically due to the interaction of Chemit cells with local neighbours in the analogue chemical domain and hysteresis effects. These high chemical states at nearest neighbours or next-nearest neighbours in different configurations lead to propagation, replication, and competition events (see Fig. 5b). Propagation and replication events occur when a high chemical state occurs at nearest and next-nearest neighbours. When two Chemits interact with each other, a competition event occurs, and for one Chemit, has a 50% survival chance. However, in some cases, multiple strong oscillations causing high chemical states to occur at nearest and next-nearest neighbours, which leads to random selection among multiple events, where a propagation, a replication, or a competition event among all the neighbours is selected randomly. Figure 5a shows a series of snapshots from the experiments showing the propagation and replication dynamics of Chemit in a 7 × 7 two-dimensional array with periodic boundary

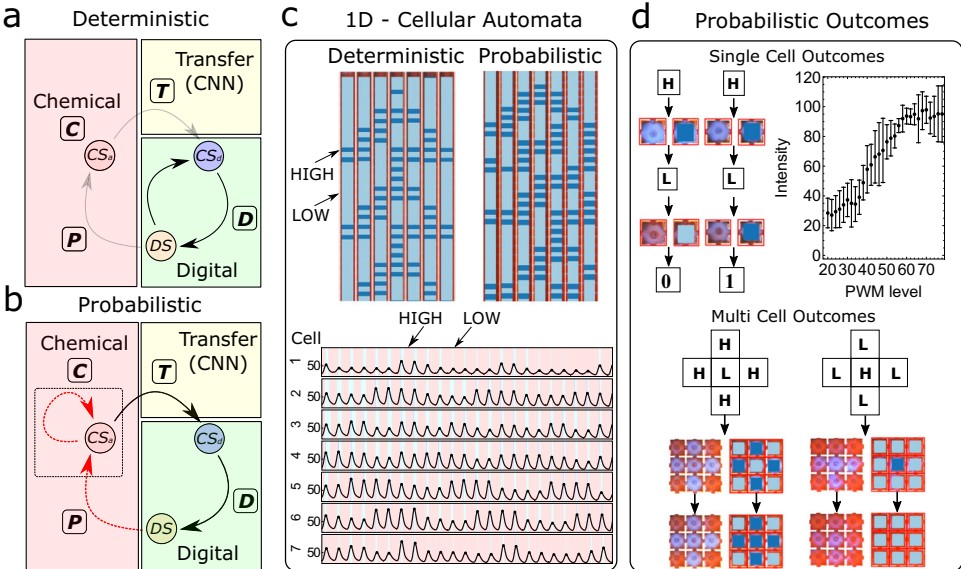

**Fig. 4 | Implementation of one-dimensional chemical cellular automata and configuration space quantification. a**, **b** Representation of hybrid electronic–chemical state machine working in deterministic and probabilistic computational modes. (Black arrow: deterministic, red arrow: probabilistic). Here, $D$, $P$, $C$ and $T$ represent digital, physical, chemical and transfer state machines and $CS_a$, $CS_d$ and represent analogue chemical state, digital chemical state, and digital state respectively. **c** Top: Implementation of elementary cellular automata (CA) (rule 30) in deterministic mode (see Supplementary Video 3) and probabilistic modes demonstrating one-to-one and many-to-one mappings (light blue: 0, blue: 1), bottom: observed oscillations with convolutional neural network (CNN) states in

the background while running CA with colours representing red, light blue and dark blue oscillatory states. **d** Top left and bottom show examples of deviations from one-to-one mapping in single and multiple cells (H and L represent high and low pulse width modulation (PWM) states). Top right shows the average peak intensity observed at different PWM levels. center points and error bars represent the mean value and standard deviation from two independent runs ($n = 2$). This data is qualitative in nature; however, it demonstrates the presence of nonlinear behaviour of observed intensities on the application of different PWM levels (see Supplementary Information Fig. 26 for more details).

conditions. The complete description of the 2D-CCA and pseudo-code is described in Supplementary Information Section 6. Over a 7 × 7 experimental array together with periodic boundary conditions, we observed the emergence of peaks in the population of the Chemits due to sudden localized replication events which later fall due to competition events within the constrained space or resource (see Fig. 5d). The total number of digitally representable high chemical states at a given step leading to Chemits dynamics are shown in Fig. 5e.

The population and the propagation dynamics of the Chemits with given control parameters are governed by the initial number of Chemits, available spatial resources, and random fluctuations and it is interesting that the system shows highly complex emergent behaviour as observed in Conway's Game of Life[25], but instantiated in a physical device in a hybrid manner. To investigate the emergent behaviour over a larger spatial scale, we developed a simple chemical probabilistic state machine based on the observed phenomenological model to simulate the dynamics of Chemits behaviour up to 150 × 150 cell array (see Supplementary Information Section 6 and Supplementary Video 5). Similar to the experimental observations, the simulations show the sudden formation of local population clusters due to fast replication and as well as annihilation due to competition events at local clusters of large populations. We observed unstable population dynamics of Chemits on an array with a smaller number of cells, however, with an increase in the number of cells the Chemits population stabilizes at different levels which depends on the available spatial resource (see Fig. 5f). We further investigated the population dynamics by varying the initial population of Chemits over a 100 × 100 array and observed convergence in the population at the steady state independent of the initial population (see Fig. 5g) and Supplementary Information Section 6 for further characterization. These simulations demonstrate a strong correlation between the global population dynamics and the local probabilistic rules emerging from digital and chemical state machines. The emergent population dynamics and

steady-state kinetics of Chemits show similarities to population behaviours observed in evolutionary biology and can be further extended towards computation operations for information processing.

## Solving combinatorial optimization problems using hybrid computational machine

Building on the dynamic feedback loop between electronic and chemical states, we implemented a hybrid electronic–chemical information processing algorithm to solve quadratic combinatorial optimization problems. Here, the idea is to utilize the probabilistic logic from the chemical analogue medium in the hybrid processor to reach the problem solution more efficiently than that of purely deterministic logic on an electronic processor. In the context of quantum adiabatic optimization, various combinatorial optimization problems such as number partitioning, satisfiability (SAT) and Hamilton cycles can be formulated as energy/cost minimization problems on an Ising lattice[29,30]. Inspired by the Ising or equivalent quadratic unconstrained binary optimization (QUBO) formulations of these problems, we implemented a hybrid electronic–chemical state machine capable of performing energy minimization using chemical states or PWM states equivalent to Ising spin variables. The generalized Hamiltonian up to a quadratic coupling is given by

$$H = h^{(0)} + \sum_{i=1}^{N} h_i^{(1)} Q_i + \sum_{i<j}^{N} h_{ij}^{(2)} Q_i Q_j \qquad (3)$$

where $Q_i$ could represent the chemical state of the cell, $h^{(0)}$ is an offset energy term, $h_i^{(1)}$ defines the self-interaction term of the spin equivalent state $Q_i$ and $h_{ij}^{(2)}$ defines the coupling between states $Q_i$ and $Q_j$. The Ising formulation of the optimization problem can be represented by a connected graph with self-interactions and pairwise couplings for Hamiltonian formulation and mapping to the chemical array (see Supplementary Information Section 7). The sign of the

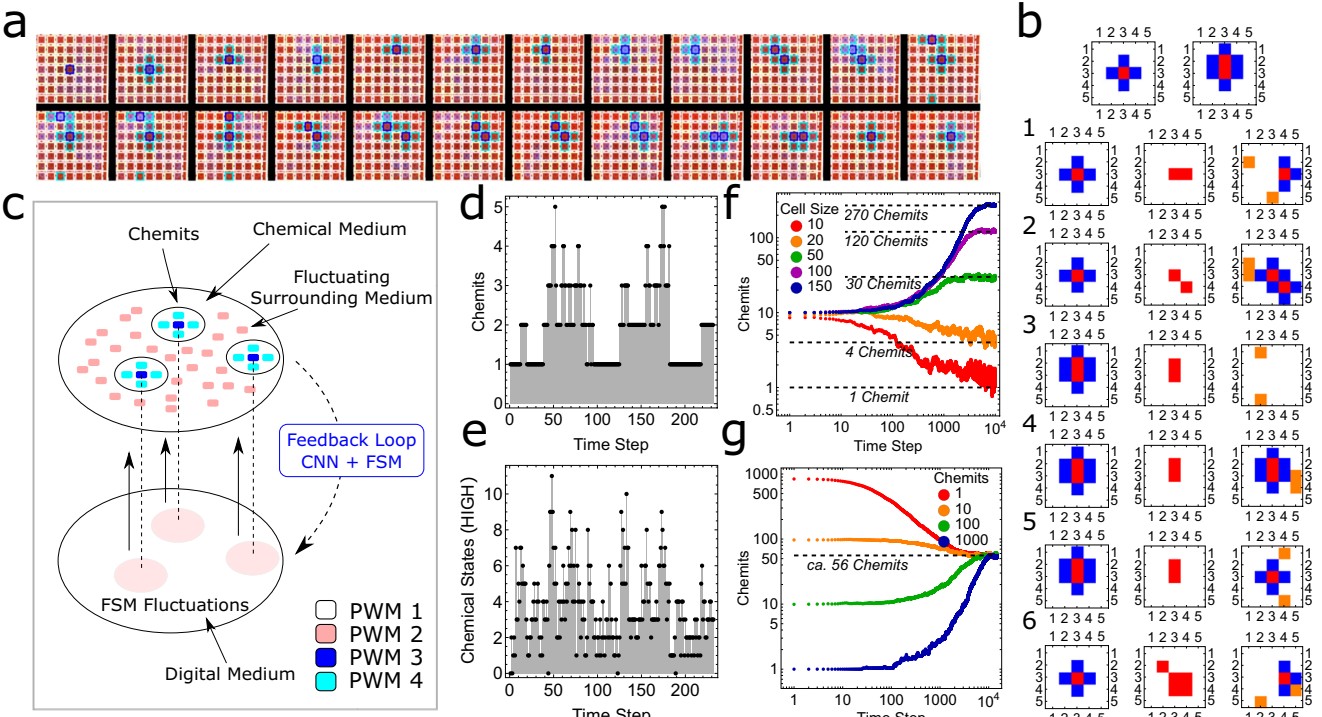

**Fig. 5 | Two-dimensional chemically instantiated probabilistic automaton.**
**a** Snapshots of the two-dimensional experimental platform showing propagation and replication events starting with a single Chemical Entity (see Supplementary Video 4). **b** The basic construct of the experimental Chemit demonstrates propagation (1), replication (2), competition (3–5), and random selection (6) between multiple events in a 5 × 5 array using the 2D-CCA state machine as a proof-of-principle. PWM states $S_0, S_1, S_2, S_3$ are shown in white, orange, red, and blue. **c** Conceptual design of closed-loop chemically instantiated probabilistic automaton, where CNN, FSM, and PWM stand for convolutional neural network, finite state machine, and pulse width modulation, respectively. **d** Population dynamics of Chemits on two-dimensional experimental step (7 × 7 cells) vs. oscillatory time steps. **e** Number of high chemical states vs. time from which Chemits are derived based on pulse width modulation (PWM) logic. **f** Average population dynamics of Chemits with different domain sizes. **g** Shows the time evolution of the average number of Chemits with different initial conditions which converge at a steady state depending on the available resource space. All simulations were run 25 times and the mean population at different time steps was estimated ($n = 25$). The numerical simulation time step is dimensionless.

coupling coefficients describes the positive (ferromagnetic type) and negative (anti-ferromagnetic type) couplings and the magnitude describes the coupling strength. In the ideal case of implementation of a computation algorithm in a hybrid system such as the proposed hybrid processor, the Ising spin equivalent chemical states should flip according to the interactions defined by local couplings. It should let the problem Hamiltonian reach the global minimum energy configuration and the solutions can be interpreted from the corresponding chemical and digital states.

As the simplest example to explore this computational problem, we have implemented an electronic–chemical hybrid state machine where the two chemical states map directly to Ising spins (−1, +1) and demonstrate the "proof-of-principle computation" through the information loop without any neighbouring interactions (see Supplementary Information Section 7 for pseudo codes and implementation details). Based on the emerging chemical states, the configurational energy is estimated from the Hamiltonian in-silico and compared with the lowest energy configuration so far and iterated until the minimal energy configuration is achieved. We show that the hybrid information processor is able to reach the energy minima successfully looping information between analogue and digital domains with higher processing occurring in the digital domain.

To increase the proportion of information processing in the chemical domain, we devised an extended approach where we utilized a combination of the digital state machine coupled with a probabilistic chemical state machine where the PWM states of the cell stirrers map directly to the Ising spin variables. The flow chart of the complete computational scheme is shown in Fig. 6a. For the pairwise

neighbouring interactions generating probabilistic outcomes, a lookup table of chemical states as shown in Fig. 6b was created and the problem was mapped on the platform (see Fig. 6c). As a result, the emergence of the new chemical states not only depends on PWM states of stirrers but also the interactions within chemical states defined by the hybrid state machine. At each step, a comparison was made between ideal states from the lookup table and the emerging chemical states and was utilized for the acceptance of the step towards energy minimization. The probabilistic outcome of the new chemical states arises from the combination of analogue and digital processing. This in turn leads to the lowest energy state due to higher connectivity in the configuration space. This improvement in efficiency will become more evident with the scaling and complexity of the problem with large local minimum configurations. By distributing the algorithm between the digital and chemical logic, we demonstrate large-scale combinatorial problems can be solved efficiently[31].

To achieve this advantage, we map the variables of the problem to the cells such that all the coupling interactions can be introduced between neighbours, see Fig. 6c which shows the mapping of a fully connected four-number partitioning problem (primary spin cell shown in blue) employing multiple instantiations of the same spins (auxiliary cells shown in red) to accomplish pairwise coupling between all the variables of the Hamiltonian. At each step, after flipping the PWM states of the cell randomly, pairwise operations to estimate the energy change were performed in a parallel approach. In the chemical decision-making step, if the emergence of the new chemical state is consistent with the lookup table, the energy change was recorded as such else was recorded with a negative sign.

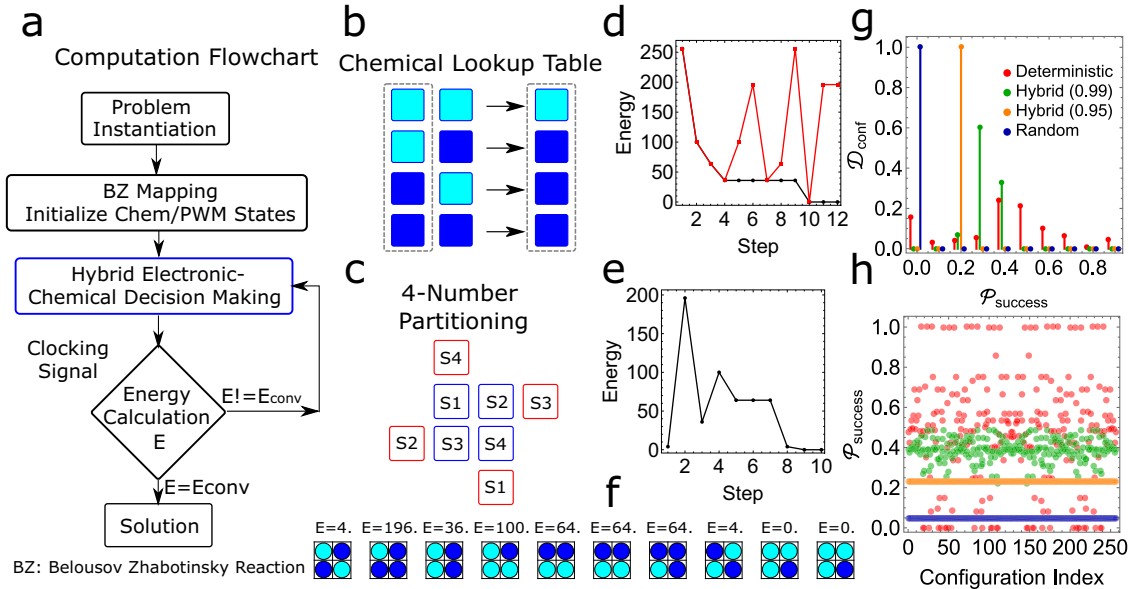

**Fig. 6 | Demonstration of hybrid computation in the chemical array. a** Flow chart describing hybrid electronic–chemical logic for solving quadratic optimization problems using a chemical array where $E_{conv}$ represents the converged energy. **b** Chemical states (CS) lookup table used for chemical decision-making based on probabilistic outcomes (light blue: $CS_i^t = 0$ and dark blue $CS_i^t = 1$). **c** Left: Mapping a 4-number partition problem on a chemical array with isolated spins with all couplings defined by neighbouring auxiliary cells. Right: Efficient mapping of 4-number partition problem on the chemical array. (Blue: principal cells, Red: auxiliary cells for spin variables). **d** and **e** Energy minimization of four-number partitioning solved using a hybrid probabilistic algorithm using two different approaches. **f** Pictorial

representation of chemical states from an experiment on energy minimization for a 4-number partition problem (light blue: $CS_i^t = 0$, dark blue: $CS_i^t = 0$), see Supplementary Video 6. **g** The distribution of initial configurations ($\mathscr{D}_{conf}$) over the success probability ($\mathscr{P}_{success}$) of solving an 8-number partition problem ($S = \{1, 3, 4, 9, 3, 5, 3, 6\}$) in pure deterministic, random and hybrid approaches. **h** Success probabilities of deterministic (index = 1.0), random (index = 0.5) and hybrid approach (index = 0.99/0.95) vs. initial configuration indices for the 8-number partition problem. The deterministic index with a value of 1.0 corresponds to a pure deterministic algorithm, 0.99/0.95 as our tuneable hybrid approach and 0.5 as a random algorithm.

The overall energy change was estimated via summation over all the pairwise spins, and the flipping was accepted or rejected similarly to the first hybrid logic scheme according to the principle of energy minimization. Once the energy associated with the current PWM states estimated from the Hamiltonian reaches the global minimum, the solution to the optimization problem is interpreted from the PWM states, see Fig. 6d and e for the solution to the four-number partitioning problem for the number set $S = \{1, 3, 4, 8\}$ using the two-hybrid algorithms. The spin state configurations of the primary spin cells for the 4-number partitioning problem using the second hybrid algorithm with the corresponding energies are shown in Fig. 6f (see Supplementary Video 6). Other examples including number partitioning, Boolean Satisfiability and Travelling Salesman Problems are demonstrated in Supplementary Information Section 7.

To investigate the influence of probabilistic logic from the chemical state machine in the hybrid approach, we quantified the algorithmic performance by defining hybrid logic as a combination of the digital algorithm coupled with analogue chemical processing. The probabilistic decision-making occurring in the chemical state machine is tuneable by selecting the PWM levels (see Fig. 4d). We formulated an 8-number partitioning problem on the number set $S = \{1, 3, 4, 9, 3, 5, 3, 6\}$ and estimated the success probability with all possible starting ($2^8 = 256$) configurations. By calculating the stationary distribution of the possible configurations, we could estimate the probability of finding the configuration with global minimum energy at different values of the deterministic index (see Supplementary Information Section 7). We observe by the reduction in the deterministic index taking advantage of the probabilistic chemical state machine; the success probability distribution shrinks leading to higher chances of finding the solution independent of the initial configuration (see Fig. 6g and h). The distribution shows for a given 8-number partition problem, the hybrid approach finds a solution with

all starting configurations however even though the deterministic algorithm shows a high probability of success, many configurations get trapped in local minima and give no solution.

## Discussion

Various unconventional computational approaches such as DNA computation, reaction-diffusion computation, optical computation, and reservoir computation utilize features of the physical systems towards computation which can have different computational principles. DNA computation performs computation using DNA sequences and requires a problem-specific design in general. Reaction-diffusion computers in the absence of a digital substrate are extremely difficult to programme and instantiating generalized problems cannot be achieved easily. Similarly, Reservoir computation in the absence of well-defined architecture could lead to strong variation in local rules within the physical substrate and therefore cannot be generalized for any computation. On the other hand, optical and quantum computers are extremely fast and show a promising future in computation, however, currently, they suffer from various scalability problems such as quantum error correction, are extremely expensive, bulky in size and manufacturing is complex. As an intermediate solution to various problems, we demonstrated a hybrid approach toward information processing and computation where an electronically programmable chemical medium is used to process information with a tuneable probabilistic logic utilizing the natural behaviour of physicochemical processes (see Supplementary Information Section 7, Qualification of hybrid electronic–chemical computation). This hybrid approach with the distribution of information between the digital and analogue chemical state machines is generic and implementable on any digitally programmable physicochemical process. Additionally, this could be further miniaturized to utilize stronger probabilistic effects at the molecular scale together with digital programmability and

instantiation. The modern computation algorithms such as machine learning (such as deep neural networks) require non-linear behaviour which is inherently present in chemical systems. This makes hybrid-computation architecture highly resource-efficient for a specific set of problems where non-linearities and probabilistic behaviour are crucial. A highly scalable architecture using hybrid computation principles as described in the manuscript could be utilized for deep learning applications (see Supplementary Information Section 8 for further discussion). The proposed architecture using BZ reaction as a programmable analogue system exhibits hybrid cellular automata rules using both digital and analogue states. We also discovered large-scale emergent dynamics of life-like entities (Chemits) resulting from the local probabilistic interactions between digital and chemical state machines. As a further extension, we also demonstrated hybrid approaches toward solving combinatorial optimization problems by minimizing Ising Hamiltonian formulation utilizing chemical states and nearest-neighbouring couplings. Importantly, we showed the distribution of information processing between the digital and the chemical domains, demonstrating that chemistry is taking an active part in the computation. Using the hybrid computational approach, larger and more complex computational problems can be mapped on the substrate. When the connectivity between the variable is larger than the possible nearest neighbour connections, multiple instantiations of the same spin can also be created to solve the problem. The use of digital components to input information into the chemical domain can be made more scalable by employing optical inputs using spatial light modulators (SLMs) or via electrochemical actuation on high-density CMOS electrode arrays. These approaches are highly scalable and fully programmable using state-of-the-art digital electronics. Similarly, the readout protocols can be substituted by a high-density CMOS sensor array to read the open circuit potential (OCP) and weak continuous flow system to stabilize cell oscillations over a long period. In that case, the use of CNN can be completely avoided reducing the role of digital computation. These input and output strategies can make the computational substrate energetically computationally efficient. The computational power in the hybrid system can be further enhanced by increasing the role of analogue chemical logic while designing new computational algorithms (see Supplementary Sections 7 and 8 for further discussions).

## Methods

### Experimental platform
The overall hybrid electronic–chemical computational platform consists of three main control domains: (a) chemical domain, (b) experimental setup, and (c) digital domain. (a) Chemical domain which consists of stock solutions required for the BZ reaction that was pumped using syringe pumps sequentially in the right proportion into the mixing chamber. The mixing chamber contains a magnetic stirrer bar that rotates at 140 RPM constantly to ensure the stock solutions are well mixed. Using another pair of syringe pumps, the reaction mixture in the mixing chamber was then transferred to the 3D-printed experimental arena with stirrers in the (b) experimental setup. In this experimental setup, the rotation of stirrers is controlled by DC motors equipped with Neodymium-based permanent magnets located at the bottom of the arena. Each motor's speed and direction can be individually addressed by the supported electronics control. The BZ chemical oscillations occurring in the experimental arena on the response of stirrer actuation were then observed and recorded by a camera. These temporal oscillatory patterns were then passed into the (c) digital domain where further information processing occurs. The oscillatory patterns were classified into three different states using a convolutional neural network (CNN). There are three different classification states, RED, LIGHT BLUE, and BLUE. These classified states were used for creating a global chemical clock over all the cells as well as the programmable chemical states (CS) for computation. The

chemical clocking logic is used over all the experiments as a sync signal for a single feedback loop step. These patterns in the given time frame were then converted into the observed chemical states using a finite state machine (FSM). This FSM reads the temporal CNN states over a given time and returns the digital CA state based on the observed oscillatory behaviour and resets. The hybrid electronic–chemical computational logic is then implemented on these chemical states using various problem-dependent state machines. Once an experiment was finished, the remaining solution was drained into the waste container using a pair of syringe pumps and the experimental arena underwent a series of rinsing and cleaning cycles to get ready for the next experiment.

### Preparation of chemical solutions
The stock solutions for the automated platforms were prepared as follows. Ferroin (0.1 M) solution was prepared by dissolving 2.78 g of ferrous sulfate heptahydrate and 5.40 g of 1,10-phenanthroline in 10 mL of deionised water. The solution was then further diluted to 0.001 M for the experiment. Sulfuric acid (1.0 M) solution was prepared by diluting 56 mL of concentrated $H_2SO_4$ to 1 L of deionised water. Potassium bromate (0.5 M) $KBrO_3$ solution was prepared by dissolving 83.5 g of $KBrO_3$ in 1 L of 1 M $H_2SO_4$. Malonic acid (1.0 M) solution was prepared by dissolving 104 g of $CH_2(COOH)_2$ in 1 L of deionised water. Deionised water PURELAB® Option-S/R 7/15 was used as the source of all water used in the experiments and preparation of stock solutions.

### State recognition using convolutional neural network
We utilized a supervised learning strategy where are large dataset was created on images of individual cells which were categorised as "red", "light blue", and "blue". Two databases were created for one- and two-dimensional experimental setups. The first database for a one-dimensional platform contains >13,000 images, and the second database from a two-dimensional platform contains >7000 images. We trained a convolutional neural network (CNN) using TensorFlow 1.X (Conv2D) based on the post-processed dataset for the classification of three distinct colour states.

### Simulations
All the simulations and further analysis were performed using Wolfram Mathematica and all the source codes are available on the repository. All the simulations regarding the dynamics of Chemits were performed 25 times at each parameter value to obtain statistically significant data.

## Data availability
Most of the source data which includes experimental and simulated data used for generating the figures in the main text and supplementary information are available on Zenodo[32] https://doi.org/10.5281/zenodo.10732131. Additional data due to large size are available upon request to the corresponding author at Lee.Cronin@glasgow.ac.uk.

## Code availability
The code used to operate the platform as various implemented simulation models are available at https://github.com/croningp/BZComputation[33] (https://doi.org/10.5281/zenodo.10723443).

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

## Acknowledgements

We would like to thank Hessam Mehr and Liam Wilbraham of the University of Glasgow for their discussions. The authors gratefully acknowledge financial support from the EPSRC (Grant Nos. EP/H024107/1, EP/I033459/1, EP/J00135X/1, EP/J015156/1, EP/K021966/1, EP/K023004/1, EP/K038885/1, EP/L015668/1, and EP/LO23652/1), the ERC (project 670467 SMART-POM), and the DARPA molecular informatics project.

## Author contributions

L.C. conceived the original idea and together with A.S. designed the project and the research plan. L.C. designed the reactor array, A.S. and M.T.-K.N. designed and built the robotic platform with help from J.M.P.G. J.M.P.G. implemented the computer vision and chemical clocking algorithms, and M.T.-K.N. created the training dataset. M.T.-K.N. and A.S. implemented the 1D-CCA. A.S., Y.J. and M.T.-K.N. implemented the 2D-CCA. A.S. and Y.J. implemented the hybrid computation and M.T.-K.N., Y.J. and A.S. performed the experiments. A.S. did the data analysis, and developed models and ran the simulations with help from Y.J. A.S. helped benchmark the system with L.C. Finally, A.S. and L.C. wrote the paper with help from the rest of the authors.

## Competing interests

The work presented here has been filed as a patent PCT/EP2023/057243.
