## [Peer Review File · Nature Communications]

A Programmable Hybrid Digital Chemical Information Processor based on the Belousov-Zhabotinsky ReactionREVIEWER COMMENTS

Reviewer #1 (Remarks to the Author):

The manuscript by Abishek Sharma et al. presents an interesting approach towards exploitation of dynamic systems in information processing. The computational systems is based on coupled layers of probabilistic chemical oscillators, digital control layer of PWM controlled stirrers and a neural network layer of image analysis. Interplay of these three layers together yields the computational system, therefore it should not be called "chemical computational device" as most of the data processing takes place in the digital part. This is rather heterotic approach, in which different computational layers are fused for the best performance. A short comment of this matter, with a reference to works of Susan Stepney would be very useful at this point.

Would it be possible to reduce the digital part as much as possible and delegate the PWM control of stirrers as well as the readout protocol to simpler, analogue layer? In this case the operation of the whole system would be solely clocked by the oscillations of the system and digital part would be eliminated - this would enrich the system, dynamics and flexibility, however its control would be much more difficult.

Clearly, I do not demand experimental verification of this concept at this stage, but a short discussion of possible development of this system would be very interesting to the whole audience of the journal.

The text may be difficult to follow for non-specialized readers, therefore suggest several small changes:

- (i) move some basic description of the single cell oscillator, results of coupling of individual cells to the main text (only a few readers will look into over 100 pages supporting material);
- (ii) clarify the generation of rules for 1D system (pages 10-11). Current version is a bit unclear. Please rewrite this section and give a clear reference to 2D system as well.
- (iii) Please suggest alternative ways of reading the output - camera and image recognition software is cool, but consumes a lot of computational resources. There must be another way(s) to read the state of the cells, maybe less sophisticated, but also much more energetically and computationally efficient.

Reviewer #2 (Remarks to the Author):

This manuscript reports about a hybrid digitally programmable chemical array and its performances in solving combinatorial optimization problems. This work is based on a programmable chemical array platform already published by the authors in Nature Communications in 2020. Although this further development has several aspects of interest, it can be considered an extension of the previously published work. The improvement of efficiency gained by combining digital with probabilistic chemical logic based on nearest neighbour interactions and hysteresis effects is a substantial improvement, however how this will affect the scalability of the computational power of the systems remains quite obscure. A benchmark evaluation against other chemical approaches or more in general, other unconventional computing approaches (for example in material reservoir computing) is also needed to assess the real impact of this kind of approach.

Response to Reviewer Comments

The reviewer comments are written in italics and responses in normal font.

Reviewer #1 (Remarks to the Author):

The manuscript by Abishek Sharma et al. presents an interesting approach towards exploitation of dynamic systems in information processing. The computational systems is based on coupled layers of probabilistic chemical oscillators, digital control layer of PWM controlled stirrers and a neural network layer of image analysis.

We thank the reviewer for the comment. The key idea we are presenting here is developing a hybrid computational logic by distributing information processing between digital and chemical systems. The digital layer defines a purely deterministic logic which is combined with natural probabilistic logic in the chemical layer. Using the described architecture, we have experimentally demonstrated three crucial features to qualify the proposed architecture for computation.

1. Chemical Clocking together with defining chemical states and their digital equivalent to create an information loop.
2. Complete programmability of the hybrid experimental system showing deterministic and probabilistic features using one- and two-dimensional Cellular Automata.
3. Solving combinatorial optimization problems using hybrid computation logic.

Using the combination of deterministic and probabilistic logic from the electronic and chemical domains, we also developed a general hybrid computational methodology which could be further applied to other physical systems as well. This has been described in SI Section 7.2 Qualification of Hybrid Digital-Chemical Computation.

Interplay of these three layers together yields the computational system, therefore it should not be called "chemical computational device" as most of the data processing takes place in the digital part.

We thank the reviewer for the suggestion, and we agree that the computation is not purely chemical but is hybrid (digital-chemical) in nature. The information processing in our system is programmable and problem specific. It can be tuned between digital and probabilistic domains depending on the problem. This problem-dependent dynamic resource allocation between digital and chemical domains is the key towards efficient computation. So, we update the "chemical computation device" as a "hybrid electronic-chemical computational device" or "hybrid computation device" in the manuscript.

This is rather heterotic approach, in which different computational layers are fused for the best performance. A short comment of this matter, with a reference to works of Susan Stepney would be very useful at this point.

We thank and agree with the reviewer that the concept and experimental design are inspired by the previous works of heterotic and reservoir computation and the works of Susan Stepney. We have added the following text and the three references to the manuscript.

“The key concept of hybrid electronic-chemical computation is inspired by the previously developed concepts of heterotic [ref. 1, ref 2, see below] as well as physical computation [ref. 3, see below] where an amalgamation of different computational systems such as digital, chemical, optical etc. are used together for efficient computation. The efficiency is gained by distributing the computational task into different computational substrates which are efficient in performing specific operations using their natural propensity.”

References:

1. Kendon, V., Sebald, A. and Stepney, S., 2015. Heterotic computing: past, present and future. *Philosophical Transactions of the Royal Society A: Mathematical, Physical and Engineering Sciences*, 373(2046), p.20140225.
2. Kendon, V., Sebald, A. and Stepney, S., 2015. Heterotic computing: exploiting hybrid computational devices. *Philosophical Transactions of the Royal Society A: Mathematical, Physical and Engineering Sciences*, 373(2046), p.20150091
3. Horsman, C., Stepney, S., Wagner, R.C. and Kendon, V., 2014. When does a physical system compute? *Proceedings of the Royal Society A: Mathematical, Physical and Engineering Sciences*, 470(2169), p.20140182.

Would it be possible to reduce the digital part as much as possible and delegate the PWM control of stirrers as well as the readout protocol to simpler, analogue layer?

We thank the reviewer for raising an important point. The key computational idea proposed here is hybrid computation, combining *the best of both worlds*: digital and chemical. The hybrid architecture proposed here is quite generic for digital-analogue systems and here we demonstrated one application by splitting deterministic computation in digital and probabilistic computation in chemical.

However, it is possible to reduce the digital component of the hybrid computational. In that case, the crucial point is the mapping of the physical problem directly on the substrate with defined computational states and completely utilising the natural propensity (time evolution) of the system for solving the problem. In principle, by fully mapping the Ising Hamiltonian for a combinatorial optimization problem on the substrate by designing both positive and negative coupling interactions using interfacial stirrers, the physical system should be able to minimize the energy (cost function) effectively reducing the digital component for the computation.

In principle, the final aim to achieve a fully “programmable chemical/probabilistic computer” requires digital processing for two key purposes. First, to instantiate a mathematical problem into the physical or chemical substrate such as initializing chemical states (as a first guess to the solution) and program the interaction coupling between the cells specific to the problem. Second, as an “amplifier” of chemical decision-making, where at each step, the chemical domains propose the search strategy for a problem and the digital layer amplifies the decision and aids the search.

Both PWM input and readout protocols could be made simpler and analogue but controlled by minimalistic digital electronics for precision.

1. The PWM input to initiate oscillations can be simplified by optical inputs utilizing Spatial Light Modulators (SLMs) spatially mapped into the chemical layer. Another possibility is to use electrochemical actuation for initiating oscillations on a high-density electrode array. Both proposed methods are analogue in nature and are highly scalable and fully programmable using state-of-the-art digital electronics.
2. Readout Protocol: The readout protocol used in the current work used a camera which reads analogue data and uses a convolutional neural network (CNN) for digital representation of the analogue state. By substituting the camera with a CMOS sensor array (electrochemical, optical) and stabilizing cell oscillations with a weak continuous flow system over all the cells, the use of CNNs could be completely avoided reducing the role of digital computation.

It this case the operation of the whole system would be solely clocked by the oscillations of the system and digital part would be eliminated - this would enrich the system, dynamics and flexibility, however its control would be much more difficult.

In principle, we have demonstrated this aspect in the manuscript. In the case of the 2d-Cellular Automata, after initializing the system, the evolution of chemical states is solely clocked by the chemical oscillations. The digital layer reads the chemical oscillations and uses a finite state logic to control the stirrers. The digital layer is also used for recognizing the chemical states using CNN to drive the Finite State Machine. Indeed, this could be further enhanced by combining chemical oscillations with photochemical control via Spatial Light Modulation (or electrochemical by high-density electrode array). In that case, chemical oscillations coupled with optical or electrochemical control will reduce the digital component still keeping the proposed hybrid computational approach intact. This could definitely increase the performance and flexibility, however, still relies on digital control (for optical and electrochemical control) for instantiating the physical problem into the computation. To keep the oscillatory system in control and to avoid drift in phase, instead of using a single global clock, multiple local clocking mechanisms can be utilized.

Clearly, I do not demand experimental verification of this concept at this stage, but a short

discussion of possible development of this system would be very interesting to the whole audience of the journal.

A short description of the minimization of the electronic component (digital) as described in the previous two questions has been updated in the final discussion section of the manuscript. This includes parts from the previous answers.

The text may be difficult to follow for non-specialized readers, therefore suggest several small changes:

We thank the reviewer for the suggestions specifically highlighting the importance of experiments performed on single-cell oscillators and coupling as well as the importance of 1D rules. They have been updated in the manuscript as suggested by the reviewer.

(i) move some basic description of the single cell oscillator, results of coupling of individual cells to the main text (only a few readers will look into over 100 pages supporting material);

A brief description of the following has been added to the manuscript:

1. Single-cell oscillator: forced and damped oscillations
2. Nearest neighbouring cell interactions: effect of coupling

(ii) clarify the generation of rules for 1D system (pages 10-11). Current version is a bit unclear. Please rewrite this section and give a clear reference to 2D system as well.

The description and the generation of the 1d Cellular Automata (as well as 2d CA) rules have been updated in the manuscript, with a more detailed description of one-dimensional CA (taken from the supplementary information), and how it has been extended to two-dimensional CA.

(iii) Please suggest alternative ways of reading the output - camera and image recognition software is cool, but consumes a lot of computational resources. There must be another way(s) to read the state of the cells, maybe less sophisticated, but also much more energetically and computationally efficient.

We thank the reviewer for the comment as this is a crucial step for the future technological development of hybrid information processing systems. The basic camera and image recognition software was utilized for its simplicity in detecting the different colour levels. The readout process can be simplified and made for efficient in two possible ways,

1. In the current setup, chemical oscillations in each cell are homogeneous and represent a single evolving chemical state. Hence, the size of each cell can be substantially minimised, and the readout of each cell could be mapped to a single RGB sensor instead of using a large section of camera pixels. This can simplify and substantially

improve the scaling as well as the energy efficiency. The use of extremely slow constant flow in the system will keep the oscillations stable over time (as discussed previously) and the requirement of CNN to detect the chemical states can be removed which can also substantially reduce energy consumption.

2. The readout of the oscillations can be made highly energy efficient and simplified by using reading oscillations of each cell electrochemically (Open Circuit Potential) on a high-density CMOS array with platinum electrodes. By designing a miniaturized interconnected cell network over an electrode array with a very weak continuous flow setup, the oscillations can be recorded directly, and the chemical states can be interpreted directly without any requirement of Convolutional Neural Networks.

A brief discussion has been added to the discussion in the manuscript.

Reviewer #2 (Remarks to the Author):

This manuscript reports about a hybrid digitally programmable chemical array and its performances in solving combinatorial optimization problems. This work is based on a programmable chemical array platform already published by the authors in Nature Communications in 2020.

We thank the reviewer for the comment. This work is inspired by the previous work published in Nature Communications (<https://www.nature.com/articles/s41467-020-15190-3>). However, we believe there are substantial differences between the previous and the current work.

This work utilizes a fundamentally different approach, where the key differences between the previous work and the current work are as follows:

1. The previous work used a 3d printed reactor array with programmable stirrers within each cell of the array. The overall coupling in the reactor array was global with no control. In the current work, the cells were carefully designed, and interfacial stirrers were introduced at the junction of the two nearest-neighbouring cells. The interfacial stirrers allow programming the coupling between the nearest neighbour cells. The local coupling introduced physical programmability in the system.
2. We introduced chemical clocking logic in our new system, which was created using a weak coupling between the nearest-neighbouring cells. This keeps chemical oscillations in phase, which is crucial for the programmability of the system and for creating the information loop.
3. As compared to previous work where the experimental platform was used as an autoencoder for pattern recognition, here we have developed a generalized hybrid computational architecture. In this architecture, using spatial local programmability, the abstract mathematical problem is instantiated into the hybrid system and the

information loops continuously between the digital and the chemical domains. Using this approach, we demonstrated programmability and computation using one- and two-dimensional Cellular Automata (BZ Game of Life) as well as the solution to combinatorial optimization problems.

4. As a computational model, we described the generalized formulation of probabilistic computation using digital and chemical states (see SI Section 7.2 Qualification of Chemical Computation). This logic can be further extended and applied to molecular-scale chemical systems as well as other physical systems which are programmable and probabilistic in nature.

Although this further development has several aspects of interest, it can be considered an extension of the previously published work.

As explained in the previous question, we believe that there are substantial differences between the previous and the current work. The key highlights shown in this work are,

1. Spatiotemporal programmability of a scalable complex chemical system utilizing its clocking logic, with tuneable probabilistic features, which was not possible in previously published work. We believe the programmability of a complex system at such a scale and accuracy has been shown for the first time.
2. This work has demonstrated a novel *physical instantiation* of two-dimensional Cellular Automata (inspired by Conway's Game of Life) showing complex dynamics of emerging life-like entities.
3. The programmability and probabilistic logic developed was used for solving combinatorial optimization problems and developing a novel hybrid computational logic involving digital-analogue systems. The key point shown here is the representation of the digital equivalent of the probabilistic chemical (analogue) state, which successfully allows the programming of complex physical systems.

The improvement of efficiency gained by combining digital with probabilistic chemical logic based on nearest neighbour interactions and hysteresis effects is a substantial improvement, however how this will affect the scalability of the computational power of the systems remains quite obscure.

We thank the reviewer to raise this important point regarding the scalability of the proposed hybrid computational architecture. The scalability of computational power has two different aspects as discussed below.

1. Mapping complex and large computational problems: Using the hybrid computational approach, larger and more complex computational problems can be mapped on the substrate. When the connectivity between the variable is larger than the possible nearest neighbour connections, multiple instantiations of the same spin can be created to solve the problem. Additionally, novel coupling architectures can be created as high-dimensional quasi space as described in SI Section. Additionally, when the

problem scales, there is no requirement of achieving global clocking, multiple local clocks can be instantiated.

2. Physical scaling of computational architecture: The proposed architecture in the manuscript is quite generic and it creates an information loop between digital and chemical domains. The highly scalable substrate can be created by miniaturizing and employing spatiotemporal optical inputs (instead of motors) using Spatial Light Modulators (SLM) or electrochemical inputs using a high-density CMOS electrode array.
3. The key aspect here is that the modern computation algorithms using Machine Learning (such as Deep Neural Networks) require non-linear behaviour which is inherently present in chemical systems. This makes hybrid-computation architecture highly resource-efficient for a specific set of problems where non-linearities and probabilistic behaviour are crucial. A highly scalable architecture using hybrid computation principles as described in the manuscript could be utilized for deep learning, see <https://www.nature.com/articles/s41586-021-04223-6> proof-of-principle example, where a hybrid system was utilized for implementing neural networks using backpropagation.

This text has been added to SI Section 8, with minimalistic modification.

A benchmark evaluation against other chemical approaches or more in general, other unconventional computing approaches (for example in materia reservoir computing) is also needed to assess the real impact of this kind of approach.

Different unconventional computational approaches use various features of the physical systems towards efficient computation, which have very different computational principles. Here, we list a few unconventional computation approaches and discuss the key differences concerning our approach.

1. DNA computation: Performing computation with DNA molecules requires problem-specific design (which uses a digital computer for synthesis and is highly expensive) and a long time to sort the correct solution among all the solutions observed. On the other hand, our hybrid computation approach might be not as parallelized as molecular scale computation, however, the architecture is fairly generic over a variety of problems, cost-effective, and solutions to the problems are directly interpretable.
2. Reaction-Diffusion computation: Reaction-Diffusion computers have shown promising potential towards computation, however in the absence of a digital substrate they are really difficult to program. Previously, Boolean logic as well as a direct solution to specific problems such as maze solving have been developed using reaction-diffusion computers, however, the architectures by themselves are not scalable due to limited programmability. Our architecture utilizes a reaction-diffusion system but makes it fully programmable and error-free (with clocking logic) such that various large-scale

mathematical problems can be mapped directly using the natural propensity of the system, instead of Boolean logic.

3. Reservoir computation: Reservoir computation architectures utilize a programmable matter for computation such as <https://doi.org/10.1038/nnano.2015.207>. Many of these architectures use “designless” architecture which has been shown effective for computation due to its highly networked structure, however, they usually require substantial time for learning the dynamics of the programmable matter and how it can be utilized efficiently for computation. Due to the lack of a well-defined structure, the local rules might differ spatially making computation in the analogue domain complicated. In our proposed architecture, the programmable matter (spatiotemporal BZ reaction) has a well-defined structural feature which limits the network connectivity, however, is locally programmable and does not drift spatially or temporally. Additionally, it does not require any initial learning due to well-defined dynamics and local programmability. The limited network is compensated by creating spatially separated multiple instances of the same variable.
4. Optical and Quantum Computation: Optical and Quantum Computers are potentially extremely fast and show a promising future in computation. However, currently, they suffer from various scalability problems such as quantum error correction, are extremely expensive, bulky in size and manufacturing is complex. On the other hand, the proposed hybrid chemical computation architecture, even if not conceptually as effective in computation as Quantum or Optical Computers, does not suffer from the issues such as scalability, error-correction. We believe the relatively lower computational performance combined with the scalability, error-free computation, cost-effectiveness is a good trade-off for future applications.

This text has been added to SI Section 8, with minimalistic modification.

REVIEWERS' COMMENTS

Reviewer #1 (Remarks to the Author):

After careful analysis of authors' responses I see that all reviewers' comments have been addressed appropriately. The qualitative comparison of computing device with other approaches is exhaustive and addresses well the problem. Therefore I recommend acceptance of the manuscript.